# Is There Indication for the Use of Biological Mesh in Cancer Patients?

**DOI:** 10.3390/jcm11206035

**Published:** 2022-10-13

**Authors:** Renato Patrone, Maddalena Leongito, Raimondo di Giacomo, Andrea Belli, Raffaele Palaia, Alfonso Amore, Vittorio Albino, Mauro Piccirillo, Carmen Cutolo, Sergio Coluccia, Aurelio Nasto, Giovanni Conzo, Anna Crispo, Vincenza Granata, Francesco Izzo

**Affiliations:** 1Dieti Department, University of Naples Federico II, 80100 Naples, Italy; 2Division of Hepatobiliary Surgical Oncology, Istituto Nazionale Tumori IRCCS Fondazione Pascale—IRCCS di Napoli, 80131 Naples, Italy; 3Division of Breast Cancer, Istituto Nazionale Tumori IRCCS Fondazione Pascale—IRCCS di Napoli, 80131 Naples, Italy; 4Melanoma and Skin Cancers Surgery Unit, Istituto Nazionale Tumori IRCCS Fondazione Pascale—IRCCS di Napoli, 80131 Naples, Italy; 5Department of Medicine, Surgery and Dentistry, University of Salerno, 84084 Salerno, Italy; 6Epidemiology and Biostatistics Unit, Istituto Nazionale Tumori IRCCS Fondazione Pascale—IRCCS di Napoli, 80131 Naples, Italy; 7Division of General Surgery, L. Curto Hospital, Polla, 84036 Salerno, Italy; 8Department of Traslational Medical Sciences, University of Campania “Luigi Vanvitelli”, 80131 Naples, Italy; 9Division of Radiology, Istituto Nazionale Tumori IRCCS Fondazione Pascale—IRCCS di Napoli, 80131 Naples, Italy

**Keywords:** ventral hernia, oncological patients, ventral hernia repair, abdominal wall, mesh, biological mesh

## Abstract

Up to 28% of all patients who undergo open surgery will develop a ventral hernia (VH) in the post-operative period. VH surgery is a debated topic in the literature, especially in oncological patients due to complex management. We searched in the surgical database of the Hepatobiliary Unit of the National Cancer Institute of Naples “G. Pascale Foundation” for all patients who underwent abdominal surgery for malignancy from January 2010 to December 2018. Our surgical approach and our choice of mesh for VH repair was planned case-by-case. We selected 57 patients that fulfilled our inclusion criteria, and we divided them into two groups: biological versus synthetic prosthesis. Anterior component separation was used in 31 patients (54.4%) vs. bridging procedure in 26 (45.6%). In 41 cases (71.9%), we used a biological mesh while a synthetic one was adopted in the remaining patients. Of our patients, 57% were male (33 male vs. 24 female) with a median age of 65 and a mean BMI of 30.8. We collected ventral hernia defects from 35 cm^2^ to 600 cm^2^ (mean 205.2 cm^2^); 30-day complications were present in 24 patients (42.1%), no 30-day mortality was reported, and 21 patients had a recurrence of pathology during study follow-up. This study confirms VH recurrence risk is not related with the type of mesh but is strongly related with BMI and type of surgery also in oncological patients.

## 1. Introduction

Despite the large diffusion of the laparoscopic approach in general and oncological abdominal surgery, the latest data available from the United States of America reported that over 2 million laparotomies are performed each year only for non-oncological patients, and in 28% of them, a ventral hernia (VH) occurs [1]. In the cancer population alone, 41% of patients developed an incisional hernia up to 2 years after resection. Even laparoscopic or laparoscopic-assisted approaches resulted in hernia formation rates of up to 23% [2]. After surgical repair, high recurrence rate was reported in the literature, with a range from 24% to 43% and a consequent health-care cost estimated around USD 3.5 billion per year [1,2,3]. 

VH repair and its correct surgical approach in all types of patients are in an evolving debate fueled by bioprosthetic device development, advances in materials, innovation in surgical techniques and multimodal approaches [4,5]. 

Moreover, in the literature, it is possible to find a great variability of nomenclature for abdominal wall planes. Varied terminology was used to refer to single anatomical planes, with consequent confusion for surgical technique description [6]. Finally, in 2019, a consensus in Delphi took stock of the situation, stipulating a list of 11 recognized terms [7].

Nonetheless, debate on VH remains a hot topic in the literature, especially in oncological patients in whom, due to the necessity of obtaining a radical resection with intricate clinical history, open approaches are often preferred to the laparoscopic one. Unfortunately, these data are obviously related with high incidence of VH and its complications. 

In fact, in oncological patients, beyond the classical risk factors for hernia development and recurrence (obesity, smoking and diabetes mellitus) there are several specific problems. Recently, Baucom et al. reported the shocking development of VH in 41% of patients surgically treated for abdominal malignancy, suggesting that cancer itself may be an additional risk factor for hernia formation [8]. Moreover, it has been proven that oncological patients are frequently associated with poor nutrition, immune deficiency, recurrent infections, advanced age and poor capacity for tissue repair. All of these factors are involved in the VH etiology [8,9]. 

Nevertheless, the management process is more complex in oncological patients because of the need to begin or restart chemotherapy as soon as possible, and it is correlated with the high possibility of multiple abdominal surgical interventions for resectable metastases. 

The literature includes many articles related to the best plane to choose or the best mesh to use in surgical VH repair, but the results and conclusions are not clear and do not provide a final response [10,11,12,13,14,15,16,17,18,19]. 

On the other hand, few articles focused on the VH impact for oncological patients and even fewer on which type of mesh would be better for these patients [20,21,22,23,24,25].

For all of these motivations, the aim of our study was to define the role of biological and synthetic implants used in cancer patients and to analyze the impact of operative choices on the VH recurrence risk.

## 2. Materials and Methods

We searched in the surgical database of the Hepatobiliary Unit of the National Cancer Institute of Naples “G. Pascale Foundation” for all patients that underwent abdominal surgery for malignancy from January 2010 to December 2018; a total of 1819 patients were found. 

Inclusion criteria for the study population were: (a)patients who had a diagnosis of abdominal malignancy;(b)patients who underwent surgical intervention in the past with radical intent;(c)patients with a diagnosis of VH on surgical site;(d)patients who underwent neoadjuvant or adjuvant chemotherapy;(e)patients who had been subject to at least 3 years of follow-up.

Exclusion criteria were: (a)patients who underwent surgical intervention for VH repair without an oncological associated surgery;(b)patients who underwent surgical intervention in the past without radical intent;(c)patients who underwent surgical intervention for abdominal malignancy without a preoperative diagnosis of VH;(d)patients with no neoadjuvant or adjuvant chemotherapy;(e)patients without a proven follow-up for at least 3 years.

Following these inclusion/exclusion criteria, 57 patients were selected and were divided into patients in whom we used synthetic prosthesis and patients in whom we used biological prosthesis.

The patients’ characteristics are listed in Table 1.

Before surgery, all selected patients signed an informed consent, and the same surgical team of the Hepato-biliary Department of our Institute performed all surgical interventions.

Considering anatomical characteristics of the VH, size of the defect, patients’ general clinical conditions, time of associated oncological surgery and radical intervention feasibility, two types of surgery were performed: bridging and anterior component separation.

Our first choice was the anterior component separation with the intent to close the defect. In these cases, meshes were used as a reinforcement. For patients in whom the fascial defect cannot be closed due to cardiological, pulmonary or surgical problems, we perform the inlay technique, which requires a bridging mesh. 

Operative time (OT), day of hospitalization (DH), 30-day complications, 30-day mortality, and radiological and clinical follow-up (at least 36 post-operative months) were reported. 

The work has been described in line with the PROCESS criteria [12].

### Statistical Methods

Continuous variables were reported as either mean ± SD or median and interquartile range (IQR) according to their distribution and tested by median with Wilcoxon signed-rank test. Categorical variables were reported as percentages and compared by using Pearson chi-squared test or Fisher’s exact test. Logistic regression was used to assess VH recurrence risk associated with a selected subset of covariates. The model was firstly built by including age, gender, BMI, type of mesh (biological or synthetic), CHT, CCI class, type of surgery, 30-day complications and the VH dimensions. Secondly, we estimated adjusted ORs and 95% confidence intervals (95% CI) through models including selected interaction variables to evaluate the potential combined effects of such clinical variables (Model A, Model B and Model C), and a goodness of fit (Mc Fadden R^2^–R^2^_MF_) was reported [26].

Median follow-up was estimated by the Kaplan–Meier approach [27].

We used the Cox models to estimate the selected hazard ratios (HRs) and 95% confidence intervals (CIs) of the regressor status [28]. HRs were adjusted by age, gender, BMI, type of mesh (biological or synthetic), CHT, CCI class, type of surgery, 30-day complications and the VH dimensions. The data were analyzed using R version 4.1.3.

## 3. Results

From January 2010 to December 2018, we treated 57 patients with incisional hernia by surgical procedure. 

Anterior component separation was used in 31 patients (54.4%) vs. bridging procedure that was used in 26 (45.6%). In 41 cases (72%), we used a biological mesh, while a synthetic one was adopted in the rest. Of our patients, 58% were male (33 male vs. 24 female) with a median age of 65 and a mean BMI of 30.8 (±2.3) kg/m^2^. 

We collected ventral hernia defects from 35 cm^2^ to 600 cm^2^ (mean: 205.2 ± 121.0 cm^2^); 30-day complications were present in 24 patients (42%), and 21 patients had a VH recurrence during study follow-up. 

A univariate analysis was carried out to assess the associations either with the type of mesh or with the VH recurrence and selected covariates. Among patients with a BMI > 30, a significant association was found with the type of mesh: in particular, in 81% (n = 13) implanted with a synthetic mesh vs. 51% with a biological one (*p* = 0.04, data not shown in table). The significant association of BMI was confirmed with VH recurrence (*p* = 0.03) and also with the type of surgery (*p* = 0.01) (see Table 2). 

Table 3 shows multivariate logistic models (Model A, Model B and Model C), which report different results according to the adjusted variables inserted in each model. The interaction variables were estimated for Models B and C (BMI x mesh; age x mesh, respectively). A higher significant risk was found for the bridging surgery (OR > 5.0, *p* < 0.025) in each model; furthermore, a significant risk was found for BMI >30 for both Model A (OR = 5.03, 95% CI 1.27–27.7) and Model C (OR = 4.8, 95% CI 1.22–23.5). Finally, Model B confirmed the significant effect of interaction between BMI and type of mesh (*p* = 0.03).

The Kaplan–Meier showed a borderline significant difference for BMI (≤30 and >30, logrank *p* = 0.054) and VH recurrence (see Figure 1); no significant differences were observed for type of mesh (logrank *p* = 0.63) (Figure 2).

A Cox proportional survival analysis was performed to evaluate the risk factors for VH recurrence. Significant HRs were found for BMI > 30 (HR = 3.48, 95% CI 1.17–10.30) and for bridging surgery (HR = 3.51, 95% CI 1.27–9.72) (Table 4).

No similar behavior was found evaluating the type of mesh (*p* = 0.63).

## 4. Discussion

Incisional hernia occurs in about 24% of patients after abdominal surgery [1]. Oncological patients had a high risk of developing incisional hernia as a result of multiple and complex surgeries required for cancer resection [2,3]. Almost all of these frail patients require surgery after perioperative chemotherapy with a consequent increasing risk of post-operative complications [29].

Despite the large diffusion of the laparoscopic approach in general and in oncological abdominal surgery, ventral hernia remains a persistent social and health problem with a high recurrence rate [8,9].

Use of synthetic or biological mesh, advances in surgical technique, better knowledge of respiratory mechanics, and better perioperative care have decreased this growing issue [30,31]. These important improvements in VH management and treatment, unfortunately, were not enough to definitely resolve the problem of recurrence.

In fact, VH is a complex problem due to many contributing causes. Biological factors, such as inflammation, granulation tissue and collagen synthesis, of paramount importance for scar and tissue regeneration, were reduced in oncological patients. In our analysis, all patients were oncological and were submitted to chemotherapy in order to have a homogeneous sample.

Other causes are related to tension force. Surgical technique had an important effect in this context. Bridging was significantly related with a high percentage of recurrence (67% vs. 33%). Maybe this result was due to a more complex surgical procedure. In fact, our first choice was the anterior component separation with the intent to close the defect. In these cases, meshes were used as a reinforcement. Only for patients in whom the fascial defect could not be closed (due to cardiological, pulmonary or surgical problems), we performed the inlay technique, requiring a bridging mesh. Obviously, radial tension on inelastic scar tissue increases the recurrence risk; paradoxically, in the bridging surgical procedure, a lower tension was observed compared with anterior component separations, but this tension works on an unclosing fascia and on the mesh without reinforcement [32,33].

Regarding tension forces, BMI has an important role. In fact, patients with BMI > 30 had a higher internal abdominal pressure due to abundant visceral fat. That increased thrust had an important impact on forces developed on the VH repaired tissue. Indeed, for every four patients with BMI > 30 treated, three had a recurrence (*p* = 0.05). Results were more impressive if analyzed in the adjusted regression models, which confirmed the role of overweight condition and the higher risk related to the surgical bridging intervention. In addition, when this sample was stratified by BMI, in the >30-group, the risk for surgical bridging increased up to a crude OR = 7.7 (95% CI 1.6–35.7 *p* = 0.009).

Our findings demonstrate a strong correlation between BMI, surgical intervention and recurrence. No significant data about the type of mesh were found; the analysis showed no statistically significant difference in univariate or regression models.

In recent years, redu-surgery has been spreading more and more for oncological patients’ treatment. Multimodal chemotherapy added to multiple surgical intervention in the same subject, gave our patients an extended expectation of life. Obviously, redu-surgery had major risk and represented a complex but safe and feasible procedure, especially in high-volume centers [34]. This new trend is strongly related with VH, and our sample represents a significant example. Furthermore, not of secondary importance is the possibility of using advanced hemostatic energetic devices that facilitate surgical possibility in reduction surgery [35,36].

Our study had some limitations. It is a retrospective study, and the sample is not very large.

Further study will be useful to understand if, in the case of redu-surgery, it will be easier for surgeons to access an abdomen that is treated with biological or synthetic mesh, and which of the two types will be safer in case of need.

## 5. Conclusions

In conclusion, biological and synthetic mesh can be used in oncological patients without a significant difference in VH recurrence rate, while a strong correlation between BMI, surgical intervention and VH recurrence risk was demonstrated.

## Figures and Tables

**Figure 1 jcm-11-06035-f001:**
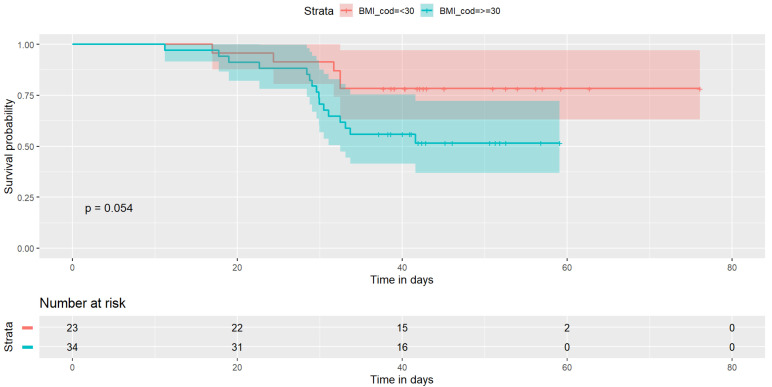
Kaplan–Meier chart stratified for BMI.

**Figure 2 jcm-11-06035-f002:**
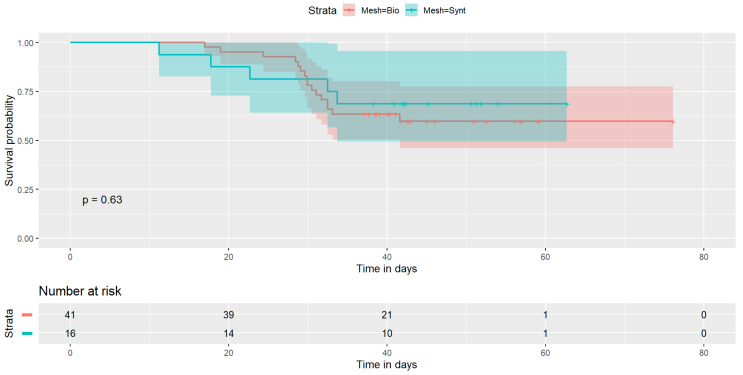
Kaplan–Meier chart stratified for type of mesh.

**Table 1 jcm-11-06035-t001:** Patient characteristics.

Variable	N (%)
Sex	
Male	33 (58)
Female	24 (42)
Age	
Mean (SD)	64.2 ± 9.8
Median (IQR)	65 (59–72)
BMI	
Mean (SD)	30.8 ± 2.3
Median (IQR)	30.5 (29.2–32.0)
Charlson index	
Mean (SD)	8.0 ± 1.1
Median (IQR)	8 (7–9)
VH Dimension	
Mean (SD)	205.2 ± 121.0
Median (IQR)	180 (140–250)
Surgery	
Bridging	26 (45.6)
Component Separation	31 (54.4)
Type of Mesh	
Bio	41 (72)
Synthetic	16 (28)
30-day complications	
No	33 (58)
Yes	24 (42)
VH Recurrence	
No	36 (63%)
Yes	21 (37%)
Primary Cancer Site	
Colo-Rectal	41 (72)
Gastric	6 (10.5%)
Ovarian	6 (10.5%)
Sarcoma	4 (7%)

Abbreviations: BMI: Body Mass Index, SD: standard deviations, IQR: interquartile range, VH: Ventral Hernia.

**Table 2 jcm-11-06035-t002:** Univariate analysis for VH recurrence.

VH Recurrence			
Variable	No N = 36	Yes N = 21	*p*-Value
Type of Mesh			0.6 ^2^
Bio	25 (69%)	16 (76%)	
Synthetic	11 (31%)	5 (24%)	
Sex			0.5 ^2^
Female	14 (39%)	10 (48%)	
Male	22 (61%)	11 (52%)	
Age (years)			>0.9 ^3^
N	36	21	
Mean (SD)	64 (10)	65 (10)	
Median (IQR)	66 (58, 71)	65 (59, 73)	
Class of age			0.6 ^2^
≤65	18 (50%)	12 (57%)	
>65	18 (50%)	9 (43%)	
BMI			**0.028** ^3^
N	36	21	
Mean (SD)	30.37 (2.18)	31.65 (2.39)	
Median (IQR)	30 (28.8, 31.4)	31.50 (30.5, 33.1)	
BMI			0.052 ^2^
<30	18 (50%)	5 (24%)	
≥30	18 (50%)	16 (76%)	
Chemotherapy			0.7 ^2^
No	19 (53%)	12 (57%)	
Yes	17 (47%)	9 (43%)	
CCI (class)			0.6 ^1^
Low (6)	5 (14%)	2 (9.5%)	
Medium (7–8)	22 (61%)	11 (52%)	
High (9–10)	9 (25%)	8 (38%)	
Surgery			**0.015** ^2^
Bridging	12 (33%)	14 (67%)	
Component separation	24 (67%)	7 (33%)	
Time of surgery			0.7 ^3^
N	36	21	
Mean (SD)	361 (100)	370 (109)	
Median (IQR)	360 (300, 420)	420 (240, 440)	
30-day complications			0.2^2^
No	23 (64.9%)	10 (47.6%)	
Yes	13 (36.1%)	11 (52.4%)	
DFS (months)			**<0.001** ^3^
N	36	21	
Mean (SD)	47 (9)	28 (7)	
Median (IQR)	43 (41, 53)	30 (24, 32)	
VH dimension (cm^2^)			0.091 ^3^
N	36	21	
Mean (SD)	181 (101)	247 (143)	
Median (IQR)	170 (98, 228)	200 (150, 300)	
VH dimension			0.13 ^2^
≤200 cm^2^	26 (72%)	11 (52%)	
>200 cm^2^	10 (28%)	10 (48%)	

^1^ Pearson’s chi-squared test; ^2^ Fisher’s exact test; ^3^ Wilcoxon rank sum test. Abbreviations—BMI: Body Mass Index, SD: standard deviations, IQR: interquartile range, VH: Ventral Hernia, CCI: Charlson Co-morbility Index, DFS: disease free survival. In bold *p*-value < 0.005.

**Table 3 jcm-11-06035-t003:** Multivariate logistic analysis of VH recurrence risk.

	Model A (R^2^_MF_ = 0.18)			Model B (R^2^_MF_ = 0.21)			Model C (R^2^_MF_ = 0.17)		
Characteristic	OR ^1^	95% CI	*p*-Value	OR ^2^	95% CI	*p*-Value	OR ^3^	95% CI	*p*-Value
Gender			0.992			0.859			0.935
Female	1^†^			1^†^			1^†^		
Male	1.01	0.27–3.90		0.89	0.24–3.41		0.95	0.25–3.75	
Type of surgery			**0.019**			**0.018**			**0.024**
Component separation	1^†^			1^†^			1^†^		
Bridging	5.14	**1.30–24.7**		5.24	**1.32–24.9**		5.04	**1.22–25.3**	
Age Class			0.239			0.212			
≤65	1^†^			1^†^					
>65	0.45	0.11–1.68		0.43	0.10–1.61				
BMI			**0.02**						**0.024**
≤30	1^†^						1^†^		
>30	5.03	**1.27–24.7**					4.80	**1.22–23.5**	
Type of Mesh			0.984						
Bio	1^†^								
Synthetic	0.98	0.17–5.39							
VH Dim.			0.729			0.389			0.736
≤200	1^†^			1^†^			1^†^		
>200	1.27	0.32–4.83		1.81	0.46–7.18		1.26	0.31–4.88	
CCI index			0.524			0.412			0.660
Low (6)	1^†^			1^†^			1^†^		
Medium (7–8)	1.09	0.16–9.76		1.42	0.18–14.7		1.04	0.15–9.44	
High (9–10)	2.50	0.27–29.2		3.62	0.35–48.3		2.00	0.22–22.1	
CHT			0.939			0.984			0.849
No	1^†^			1^†^			1^†^		
Yes	1.06	0.25–4.48		0.99	0.23–4.12		1.14	0.29–4.63	
Interaction variables									
“BMI × Mesh”						**0.027**			
If MESH is Bio				1.44	**1.10–1.99**				
If MESH is Synthetic				1.46	**1.10–2.01**				
“Age × Mesh”									0.840
If MESH is Bio							0.98	0.91–1.06	
If MESH is Synthetic							0.98	0.91–1.05	

^1^ Model with no interaction, adjusted for selected covariates. ^2^ Model with BMI and type of mesh interaction. ^3^ Model with age and type of mesh interaction. 1^†^ reference category. Abbreviations—BMI: Body Mass Index, VH: Ventral Hernia, CCI: Charlson Co-morbility Index, CHT: chemotherapy, OR: Odds Ratio, CI: Confidence Interval. In bold *p*-value < 0.005 and 95% CI with statical relevance.

**Table 4 jcm-11-06035-t004:** Cox regression model for VH recurrence.

Characteristic	HR	95% CI	*p*-Value
Sex			0.323
Female	1^†^	—	
Male	0.64	0.27–1.54	
Class of age			0.340
<=65	1^†^	—	
>65	0.64	0.26–1.61	
BMI			**0.016**
<=30	1^†^	—	
>30	3.48	**1.17–10.30**	
Mesh			0.927
Bio	1^†^	—	
Synt	0.95	0.31–2.95	
Type of surgery			**0.011**
Component separation	1^†^	—	
Bridging	3.51	**1.27–9.72**	

1^†^ reference category. Abbreviations: BMI: Body Mass Index, VH: Ventral Hernia. In bold *p*-value < 0.005 and 95% CI with statical relevance.

## Data Availability

Not applicable.

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
