# Peer review of "Is There Indication for the Use of Biological Mesh in Cancer Patients?"

_jcm, 2022, doi:10.3390/jcm11206035_

Round 1

Reviewer 1 Report

In this manuscript, Patrone R et al has showed data from a retrospective study to determine if there is a correlation between ventral hernia (VH) recurrence and use of different types of mesh. From their data analysis, they were able to conclude that there was no relation between the types of mesh used and risk of VH recurrence. Instead they demonstrated that the risk of VH recurrence was strongly dependent on patient BMI and also the type of surgery performed in oncological patients. Overall, the authors need to significantly improve the language of their paper. Few comments/ suggestions are listed below:

·         The authors need to improve the manner in which the manuscript is written. For example, in abstract it is unclear if the authors used biological mesh or whether the data that they segregated showed use of that mesh type in a particular proportion of patients.

·         In the introduction section, the authors have provided statistics from US but how does the outlook look in case of European population as data presented in the study is from patients of European descent.

·         The inclusion criteria implies abdominal malignancy but the authors have failed to clarify if the primary tumor is in the abdomen or it could be tumor of any site with metastasis present in the abdomen. The authors should explicitly state that.

·         The authors are recommended to include the study limitations in the discussion section of the manuscript.

·         In the discussion section, authors talk about redu-surgery. What does that mean?

Author Response

Dear Collegue,

first of all thank you for your time and your comment. 

As you suggest, an extensive language analysis was performed by a native. 

1) in abstract it is unclear if the authors used biological mesh or whether the data that they segregated showed use of that mesh type in a particular proportion of patients.

We selected for each patients the best available type of mesh and we chose case by case. We divided our patients in two group only after inclusion creteria selection. I hope to clearly explain our use. We add a specific sentence in the abstract (line 6-7)

2) In the introduction section, the authors have provided statistics from US but how does the outlook look in case of European population as data presented in the study is from patients of European descent.

We agree with you and we would have liked to add data about European incidence, but we can't find updated paper on this topic. We add a sentence in the introduction section (line 4-7)

3) The inclusion criteria implies abdominal malignancy but the authors have failed to clarify if the primary tumor is in the abdomen or it could be tumor of any site with metastasis present in the abdomen. The authors should explicitly state that.

Thank you. We added in the Table 1 "Primary cancer site".

4) The authors are recommended to include the study limitations in the discussion section of the manuscript.

Our limitations was included at line 24-25 of page 10 of our manuscript

5) In the discussion section, authors talk about redu-surgery. What does that mean?

We are sorry we were not clear. Redu surgery represent a new surgical treatment strategy, expecially for metastasis of colorectal cancer, in which surgeon perform a liver wedge resections so as to have a very good Future Liver Remenant. In this way surgeon will perform numerous surgical intervention on the same patients when metastasis reappear. 

If you see fit we add in the manuscript a specific sentence. 

We thank you for all your comments and we hope this review rised the quality of our manuscript and make it suitable of publications. 

Kind regards

Reviewer 2 Report

With this retrospective analysis, the authors examined the biological vs. synthetic mesh during the surgical repair (ACS and BS) of ventral hernia in 57 cancer patients. The research question of the study is clinically highly relevant. However, the study has small cohort for adequate comparison, heterogeneity, poor methodological quality, and no relevant novel finding. Finally, improving the language of the manuscript and avoiding poetic words as well as improvement of tables are recommended.

Author Response

Dear Reviewer, 

first of all thank you for your time and for your comments. 

It is true, we had a small choort and we underlined it in the text. This small sample was due to the strictly inclusion criteria adopted. For this our patients was extremely comparable with poor heterogeneity. We are so sorry you judged our manuscript with poor methodological quality. We followed all scintific steps in way to had a rigorous research manuscript. 

We try to add a new table and we added new elements in table 1. 

We performed an extensive grammar review with a native. 

We thank you for all your comments and we hope this review has increased the quality of our manuscript and makes it suitable for publications 

Kind regards

Reviewer 3 Report

In this research article, Patrone and co-Authors clearly the clinical problem regarding the development of ventral hernia (VH) in the post-operative period in patients with abdominal cancer who underwent open abdominal resection. They received biological or synthetic prosthesis. The Authors aimed to determine the role or biological or synthetic implant for cancer patients and to study the impact of the surgical choice to prevent the VH recurrence. They demonstrated that in cancer patients, VH recurrence is independent on the mesh used (biological vs synthetic) but it is strongly related to the type of resection (open resection vs laparoscopic approach) and BMI.  

The paper is well written and it flows smoothly keeping the readers on the focus. 

In the introduction, the scientific rational query is clearly presented and supported by robust literature. The method section is well detailed. The results are logically presented. 

I have a comment regarding the percentage of patients who develop VH. What is the percentage of cancer patients who underwent surgery and developed VH in the post operative period? How long after the surgery did the VH appear? 

The paper impacts a selected and specific niche of clinicians providing them a comprehensive analysis of the management process in cancer patients.

Author Response

Dear Reviewer, 

First of all thank you for your comments. 

We really appreciate your advice. 

About your questions: 

What is the percentage of cancer patients who underwent surgery and developed VH in the post operative period? 

This is a good questions. Our inclusion criteria was really stricted and we selected only few patients in way to had a better conclusions. If we analysed all surgical patients, the 7.8% developed VH after open surgery and 4.2% treated with laparoscopic or robotic approach.

How long after the surgery did the VH appear? 

Unfortunately we have no collected the time of apparence of VH for patients who not fulfilled inclusion criteria. If in your opinion is need this informations in our paper, we will perform a specific analysis.

Thank you

Round 2

Reviewer 1 Report

The authors have addressed all comments satisfactorily 

Reviewer 2 Report

The manuscript can be accepted for publication in present form.